# New Insights into X-Chromosome Reactivation during Reprogramming to Pluripotency

**DOI:** 10.3390/cells9122706

**Published:** 2020-12-17

**Authors:** Amitesh Panda, Jan J. Zylicz, Vincent Pasque

**Affiliations:** 1Laboratory of Cellular Reprogramming and Epigenetic Regulation, Department of Development and Regeneration, Leuven Stem Cell Institute, KU Leuven-University of Leuven, 3000 Leuven, Belgium; amitesh.panda@student.kuleuven.be; 2The Novo Nordisk Foundation Center for Stem Cell Biology, University of Copenhagen, 2200 Copenhagen, Denmark; jan.zylicz@sund.ku.dk

**Keywords:** epigenetics, chromatin, XCR, XCI, reprogramming, gene regulation, *Xist*, polycombs, embryo development, TAD

## Abstract

Dosage compensation between the sexes results in one X chromosome being inactivated during female mammalian development. Chromosome-wide transcriptional silencing from the inactive X chromosome (Xi) in mammalian cells is erased in a process termed X-chromosome reactivation (XCR), which has emerged as a paradigm for studying the reversal of chromatin silencing. XCR is linked with germline development and induction of naive pluripotency in the epiblast, and also takes place upon reprogramming somatic cells to induced pluripotency. XCR depends on silencing of the long non-coding RNA (lncRNA) X inactive specific transcript (*Xist)* and is linked with the erasure of chromatin silencing. Over the past years, the advent of transcriptomics and epigenomics has provided new insights into the transcriptional and chromatin dynamics with which XCR takes place. However, multiple questions remain unanswered about how chromatin and transcription related processes enable XCR. Here, we review recent work on establishing the transcriptional and chromatin kinetics of XCR, as well as discuss a model by which transcription factors mediate XCR not only via *Xist* repression, but also by direct targeting of X-linked genes.

## 1. Introduction

X-chromosome reactivation (XCR) is a developmentally regulated process by which X-chromosome inactivation (XCI) is reversed [1,2]. It originates from the evolution of sex chromosomes and dosage compensation mechanisms. The natural aneuploidy of the X chromosome in XY males [3,4] is proposed to have led to the selection of a compensation mechanism to balance X-linked gene dosage between diploid autosomes and the aneuploid X chromosome [5]. Indeed, one form of X-chromosome dosage compensation is X-chromosome upregulation (XCU), where transcriptional upregulation of the sole active X chromosome in a cell takes place [6,7,8,9]. There have been several reports supporting XCU in mammals [6,10]. There have also been reports that do not support XCU in mammals, analyses which might be explained by a lack of allele-resolution or lack of exclusion of non-expressed genes [9,11,12]. Multiple reports in the past years have confirmed that XCU occurs in placental mammals [6,13,14,15,16,17,18,19,20]. Nevertheless, while XCU solves the X-chromosome dosage problem relative to autosomes in XY cells, it potentially creates a double dose of X-linked genes in XX female cells. As a result, it is thought that eutherian mammals possess the XCI as a mechanism in which one of the two X chromosomes is randomly silenced in females to ensure gene dosage balance between the sole active upregulated X chromosome and diploid autosomes [21]. In placental mammals, XCI is random and inactivates the paternal or maternal X chromosome early in development. In turn, XCR might be a developmentally regulated process that reverses XCI in the germline. XCR has also emerged as a paradigm for studying erasure of chromatin silencing, cellular reprogramming, pluripotency, development, epigenetics and gene regulation [22,23,24,25,26,27].

Previous work has revealed a strong link between pluripotency and XCR in placental mammals [28,29,30,31]. When naive embryonic stem cells (ESCs) differentiate, random XCI is induced. In mouse, another form of XCI exists and is initiated around the four-cell stage and always takes place on the paternal X chromosome, referred to as imprinted XCI [2,28,30,31,32,33]. The paternal X chromosome then remains inactive in the trophectoderm (TE) and primitive endoderm (PE) [1,2,33]. Mice have a second form of XCR in addition to XCR in the germline, which takes place first during development where the paternal Xi undergoes XCR in the preimplantation epiblast lineage [1,2]. Paternal Xi reactivation takes place in mice but not in humans and rabbits, and therefore might be mouse-specific, perhaps linked with the need for rapid development, although many placental species remain to be examined. The strong link between the presence of two active X chromosomes (Xas) in the naive epiblast also appears in several eutherians species [1,34]. Therefore, developmentally regulated induction of pluripotency in the naive epiblast in mice and in the germline of mice or humans is associated with XCR and the presence of two active Xas [1,2,35]. As a result, naive pluripotent stem cells isolated from the naive epiblast have two Xas. Exit from naive pluripotency induces random XCI in the mouse epiblast around peri-implantation stages or in vitro upon differentiation of mouse naive ESCs [36,37,38,39,40]. Once established, chromatin silencing on the Xi is maintained despite the presence of strong *trans* factors that promote transcriptional activation of the Xa (*trans* factors are factors that have the potential to act on both alleles). Chromatin silencing in *cis* (affecting only the same allele) prevents reactivation by transcription factors and forms stable and heritable gene silencing. Although mouse pluripotent stem cells (PSCs) have been used as a key tool to study random XCI for decades, random XCI during the differentiation of human naive PSCs has only recently been reported [41]. In line with the strong link between naive pluripotency and XCR, reprogramming somatic cells to naive pluripotency induces erasure of chromatin silencing and XCR [42,43]. Partial XCR also takes place in primed human PSCs, a process termed erosion of XCI, which influences differentiation [44]. In sum, whenever naive pluripotency is induced in eutherian mammals, XCR also takes place in female cells unless an XaXa state is already pre-established.

During XCR, the long non-coding RNA (lncRNA) X inactive specific transcript (*Xist)*, a key inducer of XCI [45,46,47], is silenced, and repressive chromatin marks are lost [23,25,42,48]. *Tsix*, a lncRNA encoded antisense to *Xist,* is reactivated during XCR in mouse [23,25,42,48]. There is also X chromosome-wide remodeling of chromatin and transcription during XCR. On the Xi, all of the inactivated X-linked genes are reactivated during XCR, representing a dramatic example of erasure of facultative heterochromatin. Several events of XCR have been studied during reprograming of somatic cells into induced pluripotency, which include chromosome-wide chromatin remodeling [23,25,42,49], repression of *Xist* [23,25,42,48,49], loss of repressive chromatin marks [25,42], changes in 3D chromatin organization [49,50], loss of DNA methylation [25,51], gain of histone acetylation [48], gain of active RNA Polymerase II [25] and chromatin decompaction and changes in replication timing. XCR also leads to loss of chromosome-wide stable and heritable transcriptional silencing, as well as gain of ability to undergo random XCI [23,25,41,42,48,49,52]. Nearly all epigenetic mechanisms found for XCR are also used across the genome; hence, XCR is an excellent model to study gene regulation and epigenetics.

Temporal changes in transcription and chromatin during XCR have been the focus of several studies over the past years. In this review, we discuss recent insights into how pluripotency induction leads to XCR, reverses chromatin silencing, and erases heritable memory of transcriptional silencing. We describe an emerging model where transcription factors including members of the pluripotency gene regulatory network target multiple regulatory elements along the Xi to reverse chromatin silencing and induce transcriptional activation during reprogramming to pluripotency.

## 2. Initiation of XCI during Mouse Development and Differentiation

Initiation of XCI is a remarkable example of developmentally regulated chromosome-wide heterochromatin formation in which over 1000 genes are silenced on the Xi [53]. Although almost all genes are subject to XCI, a small proportion of X-linked genes, termed escapee genes, are not silenced and therefore maintain bi-allelic expression [54,55,56,57]. The lncRNA *Xist* has been shown to orchestrate changes in chromatin and transcription leading to the initiation, progression and maintenance of XCI [58,59,60,61]. Through RNA FISH it has been shown that *Xist* RNA expression is first partially induced on both X chromosomes upon destabilization of the pluripotency network [40], then spreads to all X chromosomes present except one [62]. *Xist* forms an RNA cloud on the Xi and spreads along the X chromosome exploiting 3D chromatin organization (Figure 1a) [62,63,64]. Intense research has been conducted to identify the regulatory elements involved in how cells sense the number of X chromosomes present in a cell (counting) and inactivate all but one X chromosome (choice) [65,66]. These efforts converged on the identification of the X inactivation center (*Xic*), a region that is necessary and sufficient for XCI [67,68,69].

*Xist* RNA triggers the recruitment of a plethora of proteins that induce, propagate and stabilize the silent chromatin state on the Xi (Figure 1b) [53,71,72,73]. The molecular events involved include histone deacetylation [71,74], gain of Polycomb dependent chromatin marks including histone H2A lysine 119 ubiquitination (H2AK119ub1) then histone H3 lysine 27 trimethylation (H3K27me3) [74,75,76,77], formation of protein condensates [78,79,80], incorporation of histone variant macroH2A [81,82] and DNA methylation [83,84,85,86,87,88]. Intense research efforts have been dedicated to understanding which elements of *Xist* RNA are responsible for the changes in chromatin and transcription described above, which has been covered in excellent recent reviews [79,89,90]. For example, the function of the E-repeat of *Xist* is delayed compared to that of the A-repeat [78,91,92]. The A-repeat appears to be the trigger for silencing while the E-repeat is important to establish stable gene silencing. In summary, several mechanisms acting in concert help to induce stable chromatin silencing on the Xi, which is maintained in somatic cells.

## 3. Maintenance of XCI

Once established, XCI is extremely stable and adopts heritable gene silencing, which can be maintained for decades. The formation of stable and heritable silenced chromatin on the Xi provides a paradigm for understanding the stability and reversibility of gene silencing. Maintaining chromatin silencing is essential for development and to prevent undesirable gene reactivation. Studies on maintenance of XCI have revealed several principles by which gene silencing is maintained. The factors and mechanisms that have been involved in maintenance of XCI include *Xist* [45,60,83,93,94,95,96,97,98,99,100,101], DNA methylation [83,84,85,88,102,103], histone deacetylation [83], Polycomb [104,105], H3K9me3 [106] and macroH2A [70,107,108], and likely include additional layers of regulation as well.

Several lines of evidence by using gene editing tools have suggested that *Xist* is not essential for maintenance of XCI. First, during XCI, there is a switch between a reversible, *Xist* dependent phase to a maintenance, *Xist*-independent phase of XCI in which *Xist* plays only a minor role [60]. Removing *Xist* expression after several days of differentiation of ESCs does not lead to XCR, arguing that other mechanisms are utilized to propagate stable and heritable gene silencing on the Xi in the absence of *Xist* expression. Second, upon *Xist* deletion in somatic cells, transcriptional silencing is largely maintained, also arguing that other mechanisms ensure maintenance of gene silencing [83,107,109]. A more recent study showed that *Xist* loss is largely tolerated in female mice [99]. Collectively, these studies have shown that *Xist* alone is not essential for maintenance of XCI.

However, several additional lines of evidence using immunofluorescence (IF) and RNA FISH have indicated that *Xist* does participate in long term maintenance of XCI, even if this role in maintenance of XCI is minor and has been underappreciated. First, several features associated with chromatin repression are lost when *Xist* is genetically deleted in somatic cells. These include loss of enrichment of Polycomb-associated repressive chromatin marks H3K27me3 and H2AK119ub1 [110], loss of repressive histone variant macroH2A on the Xi [111], loss of 3D chromatin structure increasing the probability for reacquisition of promoter-enhancer interactions [110], and chromatin opening on a small number of regions [112]. Second, long-term stability of gene silencing is compromised in the absence of *Xist* were spontaneous XCR becomes more frequent [83,93,96]. Third, when combined with other treatments such as DNA methylation and histone deacetylase (HDAC) inhibition, *Xist* deletion increases the proportion of cells that undergoes XCR [83], again suggesting a role for *Xist* in contributing to maintaining long term repression of gene silencing on the Xi and preventing erosion of XCI. Erosion of XCI in somatic cells may not manifest itself after several weeks in culture, but could be relevant in aging, although XCR in aging remains largely unclear [113]. Fourth, several studies have reported examples of partial gene-specific loss of maintenance of XCI upon *Xist* deletion in the soma [97,99]. Such examples include interesting observations in the immune system where *Xist* and heterochromatin modifications can show altered patterns in mature naive T and B cells [114,115]. How *Xist* contributes to gene-specific and cell-type specific maintenance of XCI in other contexts remains unclear. In summary, these studies suggest that *Xist* plays a minor role in maintenance of XCI in most cells, but does contribute to the long-term stability of the repressed state on the Xi in a cell-type and gene-specific manner.

A key mark for maintenance of XCI is promoter DNA methylation, which helps to maintain gene silencing especially in embryonic lineages [83,84,85,88,102,103]. Inhibiting DNA methylation with 5-Azacytidine (5Aza) induces partial reactivation [83,94,96,98,116,117,118], probably because 5Aza inefficiently inhibits DNA methylation at certain concentrations [25]. Indeed, depletion of DNMT1 by knockdown or genetically also leads to partial XCR [103,106]. Combined DNMT1 knockdown and 5Aza treatment has additive effects on DNA demethylation and XCR [25]. In addition, SMCHD1 has been implicated in DNA methylation and maintenance of about 1–10% of the genes on the Xi [84,85]. Its role in maintenance of XCI was later linked to DNA methylation independent functions [119], Polycomb repressive complex 1 (PRC1) [120], and chromatin organization [121,122]. In summary, it is thought that DNA methylation plays an important role for maintenance of XCI.

The repressive histone marks H3K9me2/3 have been shown to enrich on the Xi during XCI [106]. Depletion of the H3K9me3 transferase *Setdb1* has a very small but significant effect on maintenance of XCI [106], in line with a study in which XCI is maintained in embryos genetically depleted of H3K9 methyltransferase G9a [123]. These results indicate that H3K9me3 may have a minor role in maintenance of XCI, in line with its significant role in establishment of facultative heterochromatin [124], but other mechanisms are also involved.

Differences in the factors required to maintain XCI in distinct embryonic lineages have been reported. In the mouse TE, Polycomb group protein *Eed* maintains imprinted XCI by opposing XCR upon exit from the TE stem cell state [104,105]. Polycomb repressive complex 2 (PRC2)-mediated repression is thought to be particularly important for stable maintenance of imprinted XCI in the mouse TE because the imprinted Xi does not acquire DNA methylation as on the Xi in the soma [104,125,126,127]. A landmark study in the field showed that in *Dnmt1* knockout embryos, maintenance of XCI is unaffected in the extraembryonic endoderm while XCR takes place in the embryonic lineage [103]. A more recent study that combined genetic perturbations with single-cell RNA sequencing (scRNA-seq) in mouse embryos confirmed the role of *Eed* in maintenance of imprinted XCI in the TE [128]. However, *Eed* knockout had little impact on maintenance of XCI in the PE, indicating that distinct mechanisms maintain imprinted XCI in TE and PE lineages [128]. Given that humans have no imprinted XCI, and also have a much longer development and gestational period compared with mice, it will be interesting to define the factors and mechanisms involved in induction, maintenance and reversal of XCI in human extraembryonic lineages, which is discussed in Section 10.

One key difference between the human extraembryonic and embryonic lineages is that embryonic lineages need to maintain stable chromatin silencing for up to over 100 years, while extraembryonic lineages support embryo and fetal development for less than a year. Hence, somatic and extraembryonic lineages might use distinct mechanisms for maintenance of chromatin silencing on the Xi, and on autosomes as well. Indeed, several studies have indicated that maintenance of XCI in somatic cells depends on a plethora of distinct mechanisms that include DNA methylation.

Studies in which multiple repressive pathways are targeted individually or in combination have established that synergism between DNA methylation, histone deacetylation and *Xist* RNA together maintain gene silencing on the Xi [83]. Inhibiting DNA methylation, histone deacetylases and deleting *Xist* has a much bigger effect on XCR than any individual condition alone. Collectively, these results suggest that multiple different layers of chromatin repression act together to maintain XCI and ensure long term maintenance of XCI. 

## 4. XCR during Mouse Development

Despite the presence of most of the repressive chromatin mechanisms that we know of on the Xi, stable chromatin silencing on the Xi can be reversed. There are two forms of XCR in mouse, reversal of imprinted XCI and reversal of random XCI. During mouse preimplantation stages of embryonic development, the transcriptionally repressed paternal X (Xp) is reversed in the epiblast, whereas the extraembryonic lineages TE and PE maintain imprinted XCI for further stages of the development [1,2,26,33]. The reversal of imprinted Xp in the preimplantation epiblast is mediated by loss of *Xist* RNA, removal of repressive marks and recruitment of active marks on chromatin [1,2,129].

Previous studies have revealed a strong link between pluripotency and XCR in placental mammals [130], but not in marsupials [29]. The pluripotent cells in the preimplantation epiblast in mice and germ cells in mice and human have two Xas, while exit from pluripotency induces random XCI [41,131]. As a result, mouse PSCs derived from the preimplantation epiblast have been used extensively to study the mechanisms of XCI. 

RNA FISH and scRNA-seq experiments in early mouse embryos have shown that different genes are reactivated at different times during reversal of imprinted XCI. In preimplantation mouse embryos, XCR correlates with the expression of the pluripotency gene *Nanog* [1,2,25,132,133]. Most X-linked genes reactivate after *Xist* silencing [129,133]. However, a small category of X-linked genes reactivate before *Xist* silencing [133]. Late genes are enriched for the repressive mark H3K27me3, suggesting that this mark might oppose their reactivation [129]. These results suggested that different genes are regulated by different processes during XCR [129]. Indeed, one study suggested that the active removal of H3K27me3 by UTX may be involved, because late reactivated genes are more slowly reactivated in UTX mutant embryos [129]. It is possible that active mechanisms contribute to fast XCR in preimplantation epiblast during fast development in mice. On the other end, transcription factors NANOG and ESRRB were implicated in reactivation of early genes [129]. Furthermore, it is believed that additional mechanisms may be required to erase repressive chromatin marks during reversal of imprinted XCR. Additional studies are needed to identify the other mechanisms. Collectively, these results show that induction of pluripotency in the naive epiblast induces XCR. Accordingly, reactivation of the randomly inactivated X chromosome can also be induced in the naive epiblast of cloned embryos [52,134,135,136].

The second round of XCR in mouse occurs in primordial germ cells (PGCs) and erases random XCI [137,138,139]. The main molecular events of XCR in the germline include pluripotency-associated transcription factor activation, including *Nanog* and *Prdm14* [137], *Xist* silencing [137], H3K27me3 loss [138,140], and transcriptional reactivation [141] (Reviewed in [22,142]). More recently, it has been shown that PRDM14 mediates the removal of H3K27me3 marks from the Xi during XCR in PGCs [143]. In addition, a recent study in pig showed that *Xist* is silenced in most pre-migratory PGCs [144]. XCR begins in pre/early migrating PGCs and continues in gonadal PGCs. The kinetics of XCR is not explained by distance of genes to the *Xic* [144]. In summary, XCR takes places during epigenetic reprogramming of the germline and is linked to the expression of pluripotency genes, including transcription factors. The precise kinetics of XCR in the mouse germline remains to be defined.

## 5. XCR during Cellular Reprogramming in Mice

Several in vitro systems have been used to induce and study XCR. These include cell fusion [145,146,147,148], and nuclear transfer to mouse or frog oocytes [52,70,134] (reviewed in [22,135,149]) and somatic cell reprogramming to induced pluripotent stem cells (iPSCs) [23,25,42,48,49,150].

Reprogramming somatic cells into iPSCs erases memory of somatic cell identity and chromatin silencing [151,152]. RNA FISH studies have demonstrated that iPSC reprogramming also induces XCR [42,153] (Figure 2a). Recently, XCR has been proposed to take place in three phases: initiation, progression and completion of XCR [22]. In the initiation phase of XCR, reprogramming induces reacquisition of PRC2 enrichment on the Xi [25]. The progression phase induces reactivation of endogenous pluripotency genes [23,25,42,48,49] and include, *Tsix* reactivation [23,25,42], *Xist* silencing [23,25,42], loss of PRC2, H3K27me3 and macroH2A enrichment on the Xi [23,25,42], as well as gain of histone acetylation and active RNA Polymerase II [25,42,48], DNA demethylation [25,42,51], and transcriptional reactivation [23,25,42,48,49,150,154]. In the completion phase, stable and heritable gene silencing is reversed and the X chromosome reacquires the ability to undergo random XCI upon differentiation [42]. In summary, XCR leads to a dramatic remodeling and erasure of silenced chromatin and transcription, and takes place in distinct phases.

Multiple studies have also shown that the double dose of X-linked genes due to XCR in XX iPSCs has consequences for the molecular and cellular properties of iPSCs [51,154,155], in line with previous work in ESCs [156,157,158]. XX PSCs acquire global and sex-specific DNA hypomethylation and open chromatin landscapes [51,155,158]. Work so far suggests that reactivation of *Dusp9* leads to repression of DNMTs and induces female-specific DNA hypomethylation [51,154,155,157,158]. XX PSCs also acquire stabilization of pluripotency and exit pluripotency with delay compared with XY iPSCs, also in line with work in ESCs [51,156]. An additional elegant recent study in ESCs suggests that reactivation of *Klhl13* late in reprogramming likely stabilizes pluripotency and delays pluripotency exit in XX iPSCs [154,157,159]. Furthermore, aneuploidy of the X chromosome rapidly takes place in XX iPSCs (Figure 2b), within a couple of passages, which changes sex-specific DNA hypomethylation, open chromatin landscapes and differentiation propensity to a XY state [51,156,158]. In summary, reprogramming somatic cells to pluripotency leads to *Xist* silencing and erasure of XCI, in agreement with the strong link between pluripotency and the XaXa state, which is then responsible for inducing sex-specific molecular and cellular changes that are then lost upon acquisition of X chromosome aneuploidy.

## 6. Transcriptional Kinetics of XCR during Mouse iPSC Cell Reprogramming

Pluripotency induction in somatic cells leads to XCR [23,25,42]. Allele-resolution transcriptomic studies have been used to define the precise kinetics of XCR during iPSC reprogramming [48,49]. One study reprogrammed mouse embryonic fibroblasts and analyzed reactivation after the isolation of reprogramming intermediates using the SSEA1 marker [48]. Allele-resolution bulk RNA-seq revealed that different genes become reactivated at different times. Three categories of genes were distinguished: (1) early, (2) intermediate and (3) late reactivated genes. Early reactivated genes represent a small category of genes that become bi-allelically expressed before the entire pluripotency gene regulatory network is activated [48]. Intermediate reactivated genes represent nearly all genes on the X chromosome and their coordinated reactivation coincides with reactivation of many pluripotency transcription factors (TFs) and silencing of *Xist* [25,48]. Late reactivating genes constitute a very small class of genes that include *Klhl13*. Early reactivated genes also become highly expressed in the progression phase of XCR than in the initiation phase [48]. A more recent study used neural progenitor cells differentiated from ESCs with a *Tsix* mutation to force non-random XCI, an *Hprt*-GFP reporter, a *Nanog* reporter and an inducible reprogramming TFs cassette [49]. To define the kinetics of XCR, the neural progenitor cells were induced to reprogram to iPSCs, and analyzed using bulk RNA-seq after SSEA1 cell sorting [49]. Partial reactivation of early genes was confirmed [49].

Both studies found that early reactivated genes during iPSC reprogramming have a shorter genomic distance to escapee genes compared with intermediate and late reactivated genes [48,49]. Accordingly, the kinetics of early gene reactivation are not explained by the distance to the *Xist* locus [48,49]. Although it was suggested that early genes initiate reactivation before complete *Xist* loss [48], like in pre-implantation embryos [1,129,133], direct evidence by RNA FISH is lacking. Nevertheless, *Xist* levels are already reduced during the initiation of XCR at the time early genes become bi-allelically expressed [48,49]. Hence, it is possible that early reactivated genes may be less sensitive to *Xist* levels to maintain their silenced state during reprogramming to iPSCs. Because X-linked genes relocate to the interior of a repressive compartment during XCI, while escapee genes remain outside [160], it was proposed that early reactivated genes may be more prone to reactivation due to their 3D chromatin organization [22,48,49]. While initial Hi-C data analyses did not support or excluded this idea [48], a more recent study confirmed that early reactivated genes occupy a chromatin compartment together with escapee genes [49]. Therefore, it is possible that 3D chromatin organization could help prime genes for reactivation, but this must be experimentally tested. Recently, work in human ESCs (hESCs) found that early reactivated genes during XCR in hESCs, known as erosion of XCI (See Section 12) also starts at genes with shorter genomic distance to escapee genes [161]. Collectively, these results indicate a possible conserved link between chromatin organization and the stability and reversibility of gene silencing on the Xi, and may also be linked to the mechanisms regulating variable escape from XCI [162,163]. If confirmed and extended to autosomes, this could mean that chromatin organization might play a role in maintaining long-term heritable and stable heterochromatin silencing, and hence, stable differentiation.

Finally, two studies, one in mouse embryonic fibroblast and the other in neural progenitor cells, reached different conclusions on how fast XCR takes place during reprogramming to induced pluripotency [48,49]. The first study suggested that XCR is more gradual and hence, slower than previously recognized because early genes initiate reactivation at day 8 and late genes complete reactivation at day 12 [48]. The second study suggested that XCR can be completed within 24 h [49], like in the epiblast, suggesting that reversal of random XCI can be accelerated [129]. The difference between both studies can be explained by the criteria used to define the rate of XCR. For example, whether early reactivated and late reactivated genes are considered modifies the apparent rate of XCR. Additionally, it is also possible that the rate of XCR depends on the reprogramming system (lentiviruses versus integrated transgenes), starting cell types (embryo-derived mouse embryonic fibroblasts versus ESC-derived neural progenitor cells), and the level of reprogramming factors, in line with studies that linked the speed of reprogramming to iPSCs with starting cell states [164]. More work will be needed to determine how these parameters influence XCR. Nevertheless, all studies agree that (1) a small category of genes with genomic and epigenomic features shared with escapee genes are reactivated early and (2) most genes reactivate concomitant with reactivation of pluripotency TFs and following *Xist* silencing in a relatively short period of time. Finally, both studies use bulk measureements, and scRNA-seq will be needed to resolve the heterogeneity that is inherent to iPSC reprogramming.

## 7. Chromatin Organization of the Xi and its Dynamics during Mouse XCR

We now know that the mammalian genome is organized hierarchically (Figure 3), and the Xi adopts a unique conformation [81,165,166]. At the large scale, imaging studies have shown that each chromosome folds into individual chromosome territories. Chromosome conformation combined with sequencing techniques have shown that within each territory, active (A) and repressed (B) compartments are formed, typically hundreds to thousands of kilobase-pairs large [165,166]. At a lower scale, chromatin looping gives rise to topologically associating domains (TADs), typically tens of kilobase-pairs large, which constrain the activities of enhancers and restrict chromatin interactions [165,166]. The boundaries between TADs are enriched for CTCF and Cohesin [165,166]. 

The Xi forms two large mega-domains delimited by the *Dxz4* tandem repeat locus [81,167,168,169]. Several studies indicated a global absence or attenuation of A/B compartment structure on the Xi [73,81,121]. However, a recent study found evidence for A and B-like compartments structure present on the Xi, but attenuated due to the presence of the two mega-domains [49]. The A-like compartment on the Xi contains escapee genes and is associated with enrichment of *Xist* and H3K27me3, while the B-like compartment is enriched for heterochromatin-associated protein CBX1, also known as HP1-BETA [49]. There is also a general attenuation of TADs across the Xi, with the exception of escapee genes that display TAD-like structure [73,81,121]. In summary, the Xi adopts a unique 3D chromatin conformation characterized by mega compartments as well as attenuated A and B-like compartments and TADs. In light of the recent study that reported chromatin clutches within TADs [170], it will be interesting to define if these structures can also be found on the Xi.

Given that early reactivated genes during XCR have a shorter genomic distance to escapees [48,49] and given that escapee genes have increased A-like compartment and TAD-like structure [49], it was proposed that early reactivated genes may also have a specific 3D conformation [22,48]. Indeed, a recent transcriptomic and epigenomic study indicated that there is a relationship between 3D chromatin structure and the kinetics of XCR [49]. Early reactivated genes and escapees are enriched together in an A-like compartment on the Xi, providing evidence that early reactivated genes lie in a 3D chromatin compartment similar to escapee genes [49]. Chromatin accessibility analysis showed that, during XCI, the Xi undergoes chromosome-wide loss of accessibility, except at escapee genes [49,129]. In addition, early reactivating genes already have more accessible chromatin on the Xi before reprogramming [49]. Therefore, chromatin opening precedes transcription of early reactivating genes during the initiation of XCR [49]. Furthermore, early opening on the Xi starts within chromatin with A-like features during reprogramming, and, like for gene expression, only reaches iPSC level in the final stages of reprogramming [49]. While early reactivating genes have expression levels comparable to late reactivating genes on the Xa [49], early reactivated genes were found to be more expressed than late reactivating genes on the Xa at day-2 of reprogramming [48]. These results strongly suggest the presence of *trans* factors that are induced during reprogramming and increase the expression of early genes both on the Xa and on the Xi. Together, these results suggest that both chromatin opening and 3D chromatin organization might poise early genes for activation during iPSC reprogramming, possibly due to a combination of reduced *Xist* expression and TFs overexpression, all of which remains to be functionally tested. In summary, early reactivating genes are located adjacent to escapee genes, within an A-like compartment, and become partially open and transcriptionally reactivated early during XCR. 

3D chromatin remodeling of the *Xic*, which contains *Xist*, also takes place during XCR [49]. *Xist* silencing is key for XCR [23,25]. Early during XCR, there is a decrease in interactions within TAD-E, which contains *Xist*, *Jpx* and *Ftx*, whereas interactions between *Ftx* and *Rlim* increase [49]. *Jpx* downregulation might facilitate *Xist* repression, while *Ftx* and *Rlim* are unlikely candidates for *Xist* repression due to their reactivation kinetics [49]. Restructuring of TAD-D, which contains lncRNA *Tsix*, *Xite* and *Linx* (*Xist* repressors), occurs after XCR, suggesting that it may not be required for XCR. In summary, distinct changes in 3D-chromatin interactions take place at the *Xic* during the initiation, progression and completion of XCR.

XCR leads to the chromosome-wide reacquisition of TADs on the Xi [50]. Indeed, in situ Hi-C assays have revealed that TAD reacquisition is initiated from B-like compartments on the Xi, and is anticorrelated with *Xist* binding [49]. Different TADs are reacquired at different times during XCR, with the formation of early then late TADs. TAD formation during XCR often precedes and occurs without significant chromatin opening and gene reactivation. Early TADs do not open chromatin or reactivate genes before late TADs do. In sum, TAD formation is not sufficient for chromatin opening and gene activation, suggesting that additional mechanisms such as TF binding are necessary for erasure of chromatin silencing to take place. Two key remaining questions are (1) Is TAD formation required for chromatin opening and gene activation during XCR and (2) What is the trigger for transcriptional activation?

## 8. Mechanisms of Mouse XCR

The *Xic* has long been suspected to play a key role in XCR [67,68,69] because pluripotency TFs target repressors of XCR and activate repressors of *Xist* [28,130]. Pluripotency TFs repress *Xist* and activate *Tsix* [28,31,171]. In this way, pluripotency TFs link pluripotency with the XaXa state [130]. 

Additional mechanisms have been uncovered for erasure of Xi silencing. One study showed that the H3K27me3 histone demethylase UTX actively removes H3K27me3 on the Xi during XCR in the epiblast [129]. Whether a similar mechanism operates during iPSC reprogramming remains to be determined. It is entirely possible that the mechanisms of reversal of imprinted XCI are not all similar to those governing the reversal of random XCI, but this remains to be defined. Chromosome-wide DNA demethylation takes place during reversal of random XCI and is required but not sufficient for XCR, where combined DNA demethylation and *Xist* silencing are needed for XCR to take place [25,51]. DNA demethylation could take place via active or passive mechanisms. In favor of a model where DNA methylation is passively lost during XCR, genetic deletion of both *Tet1* and *Tet2* in addition to *Tet3* depletion does not prevent Xi DNA demethylation [25]. A key remaining question is: which additional chromatin regulators are involved in XCR and whether active or passive mechanisms are used? Candidate factors of outstanding interest for remodeling of chromatin during XCR are *Smarrc1* and *Smarca4*, which were recently identified in a screen for factors involved in XCI [150].

The proposed role of the *Xic* in inducing XCR is based on two key lines of evidence. First, *Xist* silencing is required for XCR [23,25]. Second, several pluripotency TFs bind strongly to multiple regulatory elements within the *Xic*, including the first intron of *Xist*, and the promoter of *Xist* [28]. Third, pluripotency TFs also bind and activate *Tsix* [32]. Fourth, several pluripotency-associated genes have been functionally linked to *Xist* repression [28,31,40,172]. Finally, linking pluripotency TFs to the *Xic* provides a simple and elegant mechanism linking pluripotency and XCI that also explains why loss of pluripotency triggers XCI. 

Still, there are several lines of evidence that also raise doubt whether the *Xic* is the only regulatory region required for XCR. First, deletion of *Xist* intron 1 does not prevent XCR [173]. Second, *Tsix*, a repressor of *Xist* during development [174], is not required for XCR, since its deletion does not prevent XCR [23,25]. Third, replacing the endogenous *Xist* promoter with a tetracycline inducible promoter in the presence of inducer still leads to *Xist* repression during iPSC reprogramming, indicating that the promoter of *Xist* is not required for *Xist* silencing during reprogramming [25]. Changes in organization of TAD-D and TAD-E may explain these results [49,175]. Fourth, while combined *Xist* silencing and DNA demethylation are required for XCR [25], *Xist* deletion is not sufficient for XCR in most somatic cells and reprogramming to iPSCs [25,110,112]. Fifth, the human *Xic* is not required for maintenance of XCI [109]. Therefore, *Xist* silencing is necessary but not sufficient for XCR. Together, these results indicate that *trans* factors that also target other regulatory regions, outside the *Xic* as well, are likely important for reversing transcriptional silencing on the Xi. Clearly, more work is needed to understand the mechanisms underlying XCR and to identify the *cis* regulatory regions and *trans* factors involved. 

## 9. A Possible Role for Direct Targeting of X-Linked Genes by TFs for XCR

Recent studies have shown that TFs can act as pioneer factors to induce focal chromatin opening and DNA demethylation on autosomes during somatic cell reprogramming to pluripotency [51,176]. These focal changes are seeded first focally in regulatory elements and then spread more broadly [177]. This suggests a model that may also be applicable to the Xi undergoing XCR, where pluripotency or other TFs may not only repress *Xist* but also bind across the Xi to mediate XCR. A key element of this model is that transcriptional activation requires TF binding; hence, TFs must at some point bind and activate transcription, but how this takes place during XCR remains to be determined.

Several lines of evidence suggest that direct binding of pluripotency TFs to X-linked genes within and outside the *Xic* is involved in XCR. First, in somatic cells, *Xist* deletion is not sufficient to enable TFs involved in transcriptional activation on the Xa to bind to the Xi and induce transcription [111,112]. It is likely that more than *Xist* silencing is required. Second, multiple X-linked genes outside the *Xic* are bound by pluripotency TFs in ESCs [28,32,171]. 

In this model, pluripotency TFs initially reduce *Xist* expression, enabling early genes to be reactivated by pluripotency or other TFs that bind cooperatively with pioneer factors pre-bound to open chromatin regions. Upon further *Xist* silencing, repressive marks are lost, and pluripotency TF engage in focal binding on the Xi in key regulatory regions, concomitant with focal DNA demethylation and acquisition of chromatin accessibility. Then, spreading to other regions and binding of other TFs, including both pluripotency or other TFs already engaged in transcriptional activation on the Xa, takes place. This model would explain why a small category of genes reactivate before most other genes, and would also explain why transcriptional reactivation takes place upon *Xist* silencing and activation of the entire pluripotency gene regulatory network but not upon silencing of *Xist* in somatic cells. It also provides a mechanism by which *Xist*-independent chromatin marks would be reversed on the Xi.

The model above on Xi could be wrong, and after *Xist* silencing, the mere removal of repressive marks by active or passive mechanisms could lead to binding by non-pluripotency-associated TFs. A key question is whether pluripotency TFs are involved at all in direct activation of Xi-linked gene expression during XCR, and whether they bind before or after other TFs. These questions remain to be addressed.

Recent work on transcription suggests that protein compartmentalization is likely involved in XCR. First, protein compartmentalization has just been implicated in the formation of stable gene silencing by *Xist* during XCI [78,79,80]. Therefore, if phase-separation is also found at the Xi in the maintenance phase, a mechanism by which repressive compartments are dismantled via loss of repressive chromatin protein condensates following a decrease in the concentration of critical components might be involved in XCR. Second, recent studies have also indicated that phase separation is involved in transcriptional activation through the formation of protein condensates involving the intrinsically disordered region of transcriptional activators, enhancers and promoters [178]. Therefore, XCR might also require the formation of phase separated compartments for transcription activation.

## 10. XCI in Human

Although mice have been the main model species to study XCI and XCR, recent advances have enabled to also explore dosage compensation in humans. Expression analysis, RNA FISH and IF studies have demonstrated that in 4 cell stage human embryos, unlike in mouse, XIST is expressed and forms a cloud on both X chromosome in females and on the sole X chromosome in males [35,179,180]. At this early stage, XIST does not lead to the accumulation of H3K27me3 or gene silencing, which differs from the mouse where *Xist* expression is well correlated with H3K27me3 enrichment or gene silencing. Moreover, there is no imprinted XCI in humans [181], hence, XCR does not take place in the human preimplantation epiblast. Whether the Xi of somatic cells would be reactivated after nuclear transfer in humans has not been reported, and whether XCR occurred during cloning of other female mammals including Dolly the sheep is also not clear [182]. On the one hand, these species have a pluripotent epiblast, in which one would expect two Xas. On the other hand, there is no imprinted XCI in these species and therefore whether XCR would take place in the naive epiblast when a Xi is introduced by nuclear transfer remains uncertain. 

Single-cell RNA-seq analysis has confirmed bi-allelic expression of XIST from both X chromosomes in female and from the only X chromosome in male human pre-implantation embryos [183]. Furthermore, single-cell transcriptomics also revealed that both X chromosomes in early female human embryos are active and undergo dampening, where expression of both X chromosomes become attenuated in females before random XCI is initiated [183]. In addition, models of human postimplantation development suggest that XCI takes place also during postimplantation stages, in the postimplantation epiblast and TE lineages but with delayed kinetics in the PE [35,131]. Dampening takes place in all lineages, the epiblast, TE and PE. Another recent study confirmed that both X chromosomes are active at early stages and undergo dampening [184]. However, a recent analysis questioned the dampening model and proposed that a small proportion of genes are inactivated at the 8 cell stage, followed by inactivation of other genes at the morula and blastocyst stages [185]. In addition, the function of XIST may be linked to another hominoid-specific lncRNA, XACT, that also forms an RNA cloud on the Xa [186,187]. An excellent review has been published covering general principles of XCI in human [188]. Altogether, these studies revealed that XCI is incomplete in pre-implantation human embryos and seems to proceed over the first two weeks of the human embryo development. 

## 11. XCR in the Human Germline

During human postimplantation development, genome-wide reprogramming takes place in PGCs to prepare for the next generation. Mouse experiments indicated that the Xi undergoes XCR in female PGCs [137,138,140]. scRNA-seq analysis of human PGCs showed an increase in X-to-autosome gene dosage in XX versus XY PGCs at week 4 and week 26 [189,190]. Bi-allelic expression of selected genes was also present, with a 1.6-fold increase in X-linked gene expression in female over male PGCs. It was concluded that XCR has already taken place by week 9 in human PGCs. However, whether increased X-dosage in female at week 9 represents the expression of escapee genes is not clear. Since X-linked gene expression was increased 1.6-fold in female versus male rather than 2 fold; it is possible that the male X chromosome is upregulated and the female X chromosome has erased XCU. However, more analysis is required to precisely define the kinetics of X-linked gene expression during human germline XCR. A more recent study, also using scRNA-seq, found that XCR is more heterogeneous than previously thought, where 29% PGCs have incomplete XCR by weeks 4–9 of development [191]. More work is required to further define the kinetics of XCR in the human germline. Collectively, transcriptomic studies of the human germline indicate that XCR appears to be long, taking place over 4 weeks and is asynchronous and heterogeneous. During XCR in mouse, *Xist* is silenced. However, XIST is initially expressed on the Xa in the epiblast of early human embryos before silencing is induced, raising the question of whether XIST silencing is coupled to XCR in the human female germline. Although scRNA-seq analysis showed that female PGCs express XIST much more than male PGCs [190,191], bulk RNA-seq revealed XIST expression in both male and female PGCs [192]. Moreover, XIST expression does not predict XCR [191]. IF analysis reported that H3K27me3 is enriched on the Xi in week 4 PGCs, but not week 7 PGCs where a global loss of H3K27me3 takes place [193]. A more recent study however, found a faint H3K27me3 enrichment on the Xi in a subset of PGCs of week 4 and 9, but not in week 9 and week 39.5 PGCs or in male germ cells [191]. 

It remains unclear whether XIST silencing is strictly correlated with XCR in the human germline. Collectively, the results suggest that XIST is expressed in female and male PGCs at a time when X-chromosome dosage is increased 1.6-fold in females over males. One possibility is that XIST is expressed in PGCs but does not silence, similar to pre-implantation human embryos, but different from mice. However, more work is required to answer questions such as: Does XIST form a dispersed RNA cloud as opposed to a compact RNA cloud in PGCs? Can XIST silence X-linked genes in female PGCs? Is XIST expression eventually silenced during PGCs development? scRNA-seq on human gonads as well as PGC-like cell differentiation protocols [144] could help to better understand XCR in the germline. 

## 12. XCR in Human Pluripotent Stem Cells

Work over the past years by RNA-seq and RNA FISH has shown that, with a few exceptions, most human PSCs (both hESCs and human iPSCs (hiPSCs)) have one Xi that can undergo XCR by erosion of XCI [44,179,194,195,196,197,198]. Although conventional hPSCs are in a primed state of pluripotency, they are prone to silencing XIST expression, loss of enrichment of H3K27me3 on the Xi, loss of DNA methylation and loss of transcriptional silencing on Xi [179,196]. Erosion also starts with expression of the human-specific lncRNA XACT that forms an RNA cloud on the Xa [179,199]. However, whether XACT is required for erosion is not known and its role in general remains uncertain. An excellent comprehensive analysis of RNA-seq data for hPSCs revealed that erosion was particularly present in hESCs compared with hiPSCs [198]. Erosion of XCI influences the differentiation potential of hPSCs [44]. More recently, XCR by erosion of XCI was shown to start at genes with a shorter genomic distance to escapee genes [161]. These results are in line with the shorter genomic distance of early reactivated genes during mouse iPSC reprogramming [48,49]. It is possible that conserved mechanisms are used for maintenance of XCI in human and mouse. For example, the stability and reversibility of X-linked gene silencing could depend on 3D chromatin organization as well as other chromatin and gene regulation processes, which remain to be further explored. 

## 13. XCR Following Reprogramming to Human Naive Pluripotency

Great efforts have been made to reprogram human primed PSCs as well as somatic cells to human naive PSCs, which also leads to XCR [43,200,201,202,203]. Human naive pluripotency has been covered by excellent reviews [204,205,206,207]. Below we review what is known about the dynamics of XCR during induction of human naive pluripotency. 

XCR is in fact considered one of the most stringent criteria used as a hallmark of human naive pluripotency [35,183,208]. The development of improved cell culture conditions to induce human naive pluripotency starting from primed hPSCs has provided a means to induce and study XCR in human cells [43]. A landmark study used human primed hESCs with GFP and tdTomato reporters in each allele of the X-linked MECP2 gene to show that X-linked reporter silencing is erased in naive hESCs [43]. Furthermore, primed to naive conversion was characterized by XIST reactivation in female but not in male cells, loss of promoter DNA methylation on the Xi, loss of transcriptional silencing and increased X-chromosome dosage in female compared with male cells [43]. Redifferentiation of naive cells induced non-random XCI. In summary, induction of human naive pluripotency induces XCR in female cells.

Another two studies have considerably increased our understanding of XCR during primed to naive human pluripotency conversion [179,200]. The sequence of events of XCR during primed to naive conversion includes XIST silencing [200], transcriptional reactivation [43,179,200,209], bi-allelic XACT expression [179,200,209], XIST reactivation from one allele, then XIST reactivation from two alleles in a small proportion of cells [43,200,209]. There is also gain of active Pol II and DNA demethylation [43,179,200,209]. One study reported H3K27me3 enrichment [200] and other studies reported little or no H3K27me3 enrichment under the XIST RNA foci in naive human pluripotent stem cells [180,210]. Finally, the transition also induces dampening of X-linked gene expression [200], reminiscent of early human embryos [183]. Therefore, naive cells acquire characteristics of human pre-implantation embryos, such as bi-allelic *XIST* and *XACT* expression on the Xas in a small proportion of cells. Interestingly, naive cells derived from embryos possess a much higher proportion (30% of cells) with bi-allelic *XIST* expression [200]. Moreover, H3K27me3 enrichment is not sufficient for silencing gene expression like in mouse [76,200]. Recently, another study proposed that dampening corresponds to erasure of XCU [210]. XIST reactivation is also preceded by a stage with *XIST* silencing [200]. An additional study using RNA-Seq found that XCR primarily takes place in the late stages of primed to naive conversion, in line with mouse iPSCs work [201]. A small group of genes also reactivated earlier and became mono-allelically expressed again in naive hESCs [201]. Whether these genes are closer to escapee genes remains unknown.

Early studies indicated that naive cells can undergo XCI upon differentiation, but XCI was non-random [43,200]. More recently, another study found that deriving naive cells from primed cells with reduced FGF enables the derivation of naive cells that can initiate random XCI upon redifferentiation [41]. If confirmed, this signals a revolution in the field where we will now be able to study random XCI in humans. A key objective will be to understand which regulatory mechanisms of XCI are conserved between mouse and human, and in other mammals [211].

## 14. Conclusions

The recent advent of improved developmental biology and cellular reprogramming approaches combined with genomic and gene editing technologies have provided paradigms for investigating the reversal of imprinted and random XCI. However, several unanswered questions remain (Table 1 and Figure 4).

Answering these questions will help to better understand XCR, epigenetics, gene regulation, reprogramming and embryo development. Direct binding of pluripotency TFs during XCR remains a hypothesis which will need to be further investigated. Rapid advances in single-cell genomics and multi-omics as well as reprograming and gene editing and molecular and cellular biology will help to elucidate the reversal of gene silencing during development and reprogramming.

## Figures and Tables

**Figure 1 cells-09-02706-f001:**
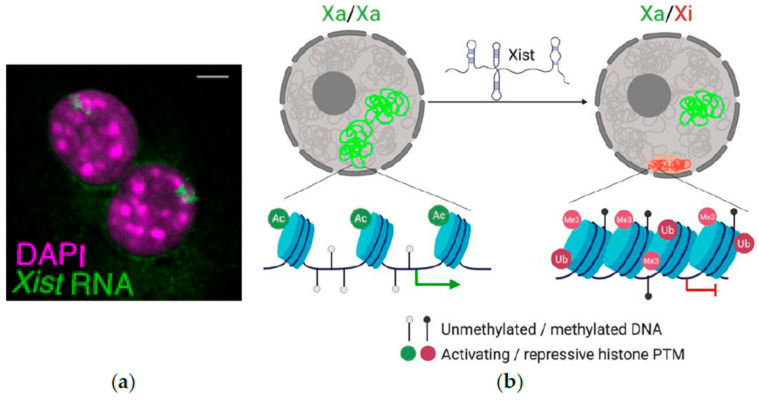
X-chromosome inactivation (XCI) in placental mammals (**a**) *Xist* RNA clouds shown by RNA fluorescence in-situ hybridization (FISH) on mouse female fibroblasts. Reproduced with permission from [70]. scale bar = 2 µm (**b**) Schematic representation of chromatin changes accompanying X-chromosome inactivation (XCI). Active X chromosomes (Xas) are shown in green. The inactive X chromosome (Xi) is shown in red. Ac: acetylation; Me3: H3K27me3; Ub: H2AK119ub1. PTM: post-translational modifications.

**Figure 2 cells-09-02706-f002:**
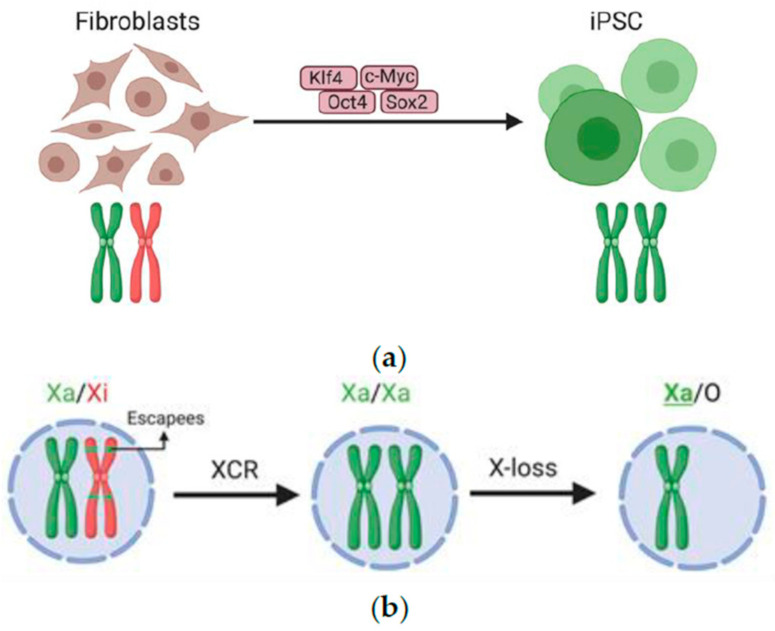
X-chromosome reactivation (XCR) during induced pluripotent stem cell (iPSC) reprogramming followed by X chromosome aneuploidy. (**a**) Roadmap showing XCR during mouse embryonic fibroblast reprogramming to iPSC by Oct4, Sox2, Klf4 and c-Myc factors; (**b**) XCR followed by rapid loss of X chromosome or X chromosome aneuploidy in XaXa iPSCs.

**Figure 3 cells-09-02706-f003:**
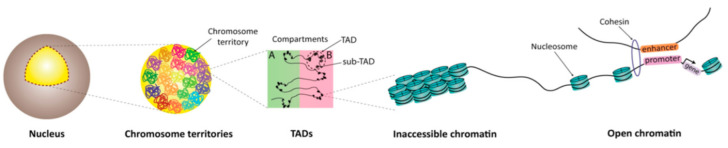
Genome organization. The mammalian genome possesses different levels of organization. From a large scale to a fine scale, chromosome territories (each in a different color), compartments A (active chromatin) and B (repressed chromatin), topologically associated domains (TADs), and nucleosomes of two homologous alleles are shown (from left to right). Middle: Compartments A and B are indicated by a green and red background, respectively. Homologous alleles are regulated independently.

**Figure 4 cells-09-02706-f004:**
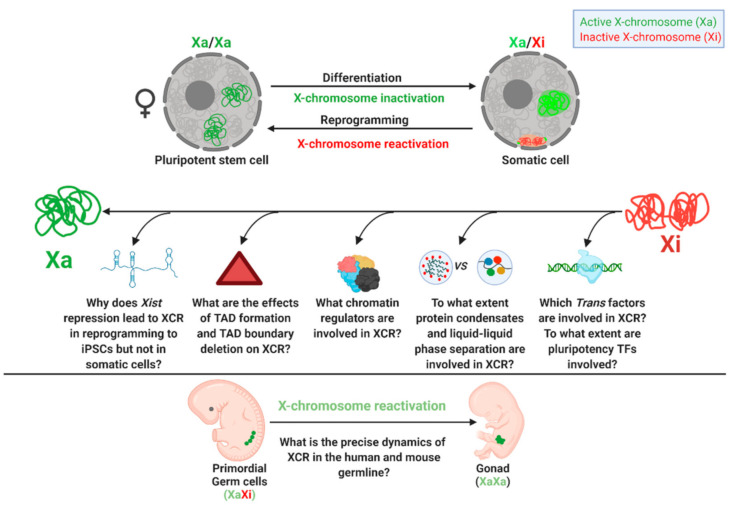
Gaps in our knowledge of XCR. Several unanswered questions during X-chromosome reactivations during reprogramming to pluripotent stem cell from somatic cell and also during germline development are depicted.

**Table 1 cells-09-02706-t001:** Key questions raised in this review.

Which *trans* factor are involved in XCR? To what extent are pluripotency TFs involved? What chromatin regulators are involved in XCR? How are regulatory regions outside the *Xic* targeted for erasure of repressive marks and acquisition of active marks? Are there pioneer TFs that target Xi-linked genes for reactivation during XCR? Is TAD formation required for chromatin opening and gene activation during XCR What are the effects of TAD boundary deletion on the kinetics of transcriptional activation during XCR? Why does *Xist* repression lead to XCR in reprogramming to iPSCs but not in somatic cells? By which mechanisms are repressive chromatin marks removed during XCR? To what extent are protein condensates and phase separation involved in XCR? What are the precise dynamics of XCR in the mouse and human germline? What triggers transcriptional reactivation during XCR? What factors are required for the initiation, maintenance and reversal of XCI in the human TE?

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
