# Peer review of "New Insights into X-Chromosome Reactivation during Reprogramming to Pluripotency"

_cells, 2020, doi:10.3390/cells9122706_

Round 1

Reviewer 1 Report

In this interesting review paper Panda and Pasque provide a comprehensive overview of the X-reactivation phenomenon. It takes both classical as well as the very latest literature into account, provides balanced interpretation of the different studies and proposes interesting new hypotheses. I enjoyed reading it. Below I list the inaccuracies/mistakes, which should be corrected before publication.

Major Comments

  1. line 40-43: “As a result, it is thought that metatherian and eutherian mammals evolved subsequently to XCI as a mechanism in which one of the two X chromosomes is randomly silenced in females to ensure gene dosage balance between the sole active upregulated X chromosome and diploid autosomes [18].”

This is incorrect for metatherians (marsupials), which do not show random, but rather imprinted XCI. Also remove “to” in front of XCI.

  1. line 49-50: “Previous work has revealed a strong link between pluripotency and XCI in placental mammals, but not in monotremes [25–28].”

This sentence should be corrected. There is a link between XCR (not XCI) and pluripotency in placental mammals. Marsupials (e.g. Monodelphis domestica) indeed do not show this link. The third mammalian clade, monotremes (egg laying mammals, e.g. platypus) have an entirely different sex chromosome composition (5X and 5Y chromosomes) and to my knowledge the dosage compensation mechanism in them is not clearly know, so they should not be mentioned here, also as the references cited (25-28) do not discuss the link between pluripotency and XCI/XCR in monotremes.

  1. line 56-57: “Paternal Xi reactivation does not occur in other placental mammals and appears to be a mouse 
invention, perhaps linked with the need for rapid development.” 
I would tone this statement down, as not many placental species (besides mouse, human and rabbit) have been yet extensively studied for random versus imprinted XCI and XCR in the ICM. Paternal Xi reactivation might be therefore also present in other placental mammals and not exclusively in mice.

  1. line 64-65: “Once established, XCI is maintained despite the presence of the trans factors, that 
underlie expression of the Xa.”

This meaning of this sentence is a bit unclear. Please rephrase or explain better.

  1. line 66-70: “Although mouse pluripotent stem cells (PSCs) 
have been used as a key tool to study random XCI for decades, random XCI in human naive PSCs 
has only recently been reported [38]. In line with the strong link between naive pluripotency and 
XCI, reprogramming somatic cells to naive pluripotency induces erasure of chromatin silencing and 
XCR [39,40]. “

Again, there is a link between XCR (not XCI) and naïve pluripotency and XCI can be studied during differentiation of PSC not in naïve PSCs. Please correct this as it might be confusing/misleading for a non-expert reader.

  1. line 82-83: “and most likely loss of nuclear periphery localization”.

This should be removed, as the active Xa is also known to be found at the nuclear periphery (even more so than the Xi). See: Teller K, Illner D, Thamm S, Casas-Delucchi CS, Versteeg R, Indemans M, Cremer T, Cremer M. 2011. A top-down analysis of Xa- and Xi-territories reveals differences of higher order structure at ≥20 Mb genomic length scales. Nucleus 2, 465–477. (10.4161/nucl.2.5.17862) 

  1. line 131: “Several lines of evidence have suggested that Xist is not important for maintenance of XCI.”

While it is true that Xist is not essential, it is not correct that it is not important for maintenance of XCI (as explained in the next paragraph of the manuscript). Xist is one of several redundant layers keeping the Xi silent, and when for example DNA-methylation is removed, Xist becomes important to keep the silent state (e.g. References 77, 85, 87, 22). According to that logic, one could make the argument that no single layer of silencing (including DNA-methylation) is essential for maintenance of XCI, as perturbation of each single one individually is not sufficient for XCR. I would therefore rephrase this sentence, which will then better match the following paragraph.

  1. line 143-145: “These include loss of enrichment of repressive histone variant 
macroH2A on the Xi [102], loss of 3D chromatin structure increasing the probability for reacquisition of promoter-enhancer interactions [103], and chromatin opening on a small number of regions [104].”

Importantly, also Polycomb and its associated chromatin marks are rapidly lost after Xist depletion. Please mention this.

  1. line 205-207: During mouse peri-implantation stages of embryonic development, the transcriptionally 
repressed paternal X (Xp) is reversed in the ICM, whereas the extra-embryonic lineages maintain 
imprinted XCI for further stages of the development [1,2,23,30].”

Please be more specific here and in other parts of the manuscript where XCR in the ICM is mentioned. XCR occurs only in the blastocyst epiblast and not in the whole ICM as the PE, which is part of the ICM, does not undergo XCR.

  1. line 384-386: “In summary, early reactivating genes have open chromatin on the Xi and share several chromatin 
organization features with escapee genes, which might poise them for early reactivation during 
XCR. “

Early genes are not yet completely open on the Xi. I would rephrase this and say instead something like: “In summary, early reactivating genes are located adjacent to open chromatin of escapee genes, poising them to become opened early during reprogramming, which is followed by their partial transcriptional reactivation early during XCR.
”Ftx might 
facilitate Xist repression, while Ftx and Rlim are unlikely candidates for Xist repression due to their reactivation kinetics [46].“

Please correct to: “Jpx downregulation might facilitate Xist repression, while Ftx and Rlim are unlikely candidates for Xist repression due to their reactivation kinetics [46].”

  1. line 421-423: “Candidate factors of outstanding interest for remodeling of during XCR are Smarrc1 and Smarca4, which were recently identified in a screen for 
factors

involved in XCI [164].

Please correct the reference – this one is incorrect.

Minor Comments

  1. “X chromosome” is correctly spelled without a hyphen, however “X-chromosome inactivation”, “X-chromosome reactivation”, “X-chromosome upregulation” etc. should be spelled with a hyphen (not done in this manuscript starting with the title). Please correct this throughout the manuscript.

  1. line 65: transcription factors (not transcriptions factors).

  1. line 86: Full stop missing at end of sentence.

  1. line 216-217: “In pre0implantation mouse 
embryos, XCR correlates with the expression of the pluripotency gene Nanog [22,124].

Correct the word “preimplantation” and add references 1, 20 and 125, which show XCR in preimplantation embryos in Nanog+ cells (reference 22 shows it during iPSC reprogramming but not in preimplantation embryos).

  1. line 248: erase not erases

  1. line 582: pluripotency not ply potency 

Author Response

Point-by-point Response to the Reviewers Comments:

Manuscript ID: cells-1003543

We would like to thank the reviewers for their constructive comments and helpful suggestions. We have been able to address all of the points raised in our revised manuscript. We feel that these changes have greatly improved our review. Below we provide a point-by-point rebuttal to all of the comments made by the reviewers.

Reviewer 1 expressed interest in the review, stating that it "provides balanced interpretation of the different studies and proposes interesting new hypotheses". The reviewer enjoyed reading the piece and made a number of excellent and well-informed suggestions aimed at improving the manuscript, which we have now addressed.

Comments and Suggestions for Authors

In this interesting review paper Panda and Pasque provide a comprehensive overview of the X-reactivation phenomenon. It takes both classical as well as the very latest literature into account, provides balanced interpretation of the different studies and proposes interesting new hypotheses. I enjoyed reading it. Below I list the inaccuracies/mistakes, which should be corrected before publication.

We are grateful to reviewer 1 for well-informed comments that were of considerable value in improving our manuscript. We also thank reviewer 1 for agreeing to reveal her/his identity upon completion of the review process. Both Panda and Pasque have made different guesses about the identity of reviewer 1 and we look forward to seeing if we got it right or not. In the meantime, we have addressed the comments, as detailed below.

Major Comments

  1. line 40-43: “As a result, it is thought that metatherian and eutherian mammals evolved subsequently to XCI as a mechanism in which one of the two X chromosomes is randomly silenced in females to ensure gene dosage balance between the sole active upregulated X chromosome and diploid autosomes [18].”

This is incorrect for metatherians (marsupials), which do not show random, but rather imprinted XCI. Also remove “to” in front of XCI.

We agree with the reviewer. "metatherian" and "to" have now been removed from the sentence, line 48-51.

  1. line 49-50: “Previous work has revealed a strong link between pluripotency and XCI in placental mammals, but not in monotremes [25–28].”

This sentence should be corrected. There is a link between XCR (not XCI) and pluripotency in placental mammals. Marsupials (e.g. Monodelphis domestica) indeed do not show this link. The third mammalian clade, monotremes (egg laying mammals, e.g. platypus) have an entirely different sex chromosome composition (5X and 5Y chromosomes) and to my knowledge the dosage compensation mechanism in them is not clearly know, so they should not be mentioned here, also as the references cited (25-28) do not discuss the link between pluripotency and XCI/XCR in monotremes.

We agree with the reviewer. XCI has been switched to XCR (line 57). We have now also deleted "but not in monotremes" from the sentence (line 58).

  1. line 56-57: “Paternal Xi reactivation does not occur in other placental mammals and appears to be a mouse 
invention, perhaps linked with the need for rapid development.” 
I would tone this statement down, as not many placental species (besides mouse, human and rabbit) have been yet extensively studied for random versus imprinted XCI and XCR in the ICM. Paternal Xi reactivation might be therefore also present in other placental mammals and not exclusively in mice.

We fully agree with the reviewer. As suggested, we have toned down the sentence: Lines 64-67.

"Paternal Xi reactivation takes place in mouse but not in humans and rabbit, and therefore might be a mouse invention, perhaps linked with the need for rapid development, although many placental species remain to be examined. "

  1. line 64-65: “Once established, XCI is maintained despite the presence of the trans factors, that 
underlie expression of the Xa.”

This meaning of this sentence is a bit unclear. Please rephrase or explain better.

 The sentence has now been rephrased. " Once established, chromatin silencing on the Xi is maintained despite the presence of strong trans factors that promote transcriptional activation of the Xa (trans factors are factors that have the potential to act on both alleles)", lines 78-81.

  1. line 66-70: “Although mouse pluripotent stem cells (PSCs) 
have been used as a key tool to study random XCI for decades, random XCI in human naive PSCs 
has only recently been reported [38]. In line with the strong link between naive pluripotency and 
XCI, reprogramming somatic cells to naive pluripotency induces erasure of chromatin silencing and 
XCR [39,40]. 

Again, there is a link between XCR (not XCI) and naïve pluripotency and XCI can be studied during differentiation of PSC not in naïve PSCs. Please correct this as it might be confusing/misleading for a non-expert reader.

As requested, this has been corrected, and "XCI" was replaced by "XCR", lines 85-87.

  1. line 82-83: “and most likely loss of nuclear periphery localization”.

This should be removed, as the active Xa is also known to be found at the nuclear periphery (even more so than the Xi). See: Teller K, Illner D, Thamm S, Casas-Delucchi CS, Versteeg R, Indemans M, Cremer T, Cremer M. 2011. A top-down analysis of Xa- and Xi-territories reveals differences of higher order structure at ≥20 Mb genomic length scales. Nucleus 2, 465–477. (10.4161/nucl.2.5.17862) 

The statement has now been removed, as suggested.

  1. line 131: “Several lines of evidence have suggested that Xist is not important for maintenance of XCI.”

While it is true that Xist is not essential, it is not correct that it is not important for maintenance of XCI (as explained in the next paragraph of the manuscript). Xist is one of several redundant layers keeping the Xi silent, and when for example DNA-methylation is removed, Xist becomes important to keep the silent state (e.g. References 77, 85, 87, 22). According to that logic, one could make the argument that no single layer of silencing (including DNA-methylation) is essential for maintenance of XCI, as perturbation of each single one individually is not sufficient for XCR. I would therefore rephrase this sentence, which will then better match the following paragraph.

We agree with the reviewer and have rephrased the sentence, as suggested.  " Collectively, these studies have shown that Xist alone is not essential for maintenance of XCI." Lines 165-166.

  1. line 143-145: “These include loss of enrichment of repressive histone variant 
macroH2A on the Xi [102], loss of 3D chromatin structure increasing the probability for reacquisition of promoter-enhancer interactions [103], and chromatin opening on a small number of regions [104].”

Importantly, also Polycomb and its associated chromatin marks are rapidly lost after Xist depletion. Please mention this.

We agree that Polycomb-associated marks are lost after Xist deletion during the maintenance phase of X inactivation. We have now mentioned loss of Polycomb-associated chromatin marks, as suggested, Lines 170-174.  

  1. line 205-207: During mouse peri-implantation stages of embryonic development, the transcriptionally 
repressed paternal X (Xp) is reversed in the ICM, whereas the extra-embryonic lineages maintain 
imprinted XCI for further stages of the development [1,2,23,30].”

Please be more specific here and in other parts of the manuscript where XCR in the ICM is mentioned. XCR occurs only in the blastocyst epiblast and not in the whole ICM as the PE, which is part of the ICM, does not undergo XCR.

We fully agree with the reviewer on this. XCR takes place in the naive epiblast no the whole ICM. We have corrected this throughout the manuscript, lines 239-242.

  1. line 384-386: “In summary, early reactivating genes have open chromatin on the Xi and share several chromatin
organization features with escapee genes, which might poise them for early reactivation during 

Early genes are not yet completely open on the Xi. I would rephrase this and say instead something like: “In summary, early reactivating genes are located adjacent to open chromatin of escapee genes, poising them to become opened early during reprogramming, which is followed by their partial transcriptional reactivation early during XCR.
”Ftx might 
facilitate Xist repression, while Ftx and Rlim are unlikely candidates for Xist repression due to their reactivation kinetics [46].“

We agree that early genes are not yet completelly open on Xi in fibroblasts. As suggested, we have rephrased this sentence which now reads " In summary, early reactivating genes are located adjacent to escapee genes, within an A-like compartment, and become partially open and transcriptionally reactivated early during XCR.", lines 423-425.

Please correct to: “Jpx downregulation might facilitate Xist repression, while Ftx and Rlim are unlikely candidates for Xist repression due to their reactivation kinetics [46].”

As suggested, this has now been corrected, lines 432-433.

  1. line 421-423: “Candidate factors of outstanding interest for remodeling of during XCR are Smarrc1 and Smarca4, which were recently identified in a screen for 
factors

involved in XCI [164].

Please correct the reference – this one is incorrect.

We agree with the reviewer. Thank you for pointing this out. We meant to cite Kinery et al. BioRxiv 2019 and have now made the change.

Minor Comments

  1. “X chromosome” is correctly spelled without a hyphen, however “X-chromosome inactivation”, “X-chromosome reactivation”, “X-chromosome upregulation” etc. should be spelled with a hyphen (not done in this manuscript starting with the title). Please correct this throughout the manuscript.

 We made the changes in formatting of all such instances, as suggested by the reviewer.

  1. line 65: transcription factors (not transcriptions factors).

 The typographic error has been corrected.

  1. line 86: Full stop missing at end of sentence.

 The correction has been made.

  1. line 216-217: “In pre0implantation mouse 
embryos, XCR correlates with the expression of the pluripotency gene Nanog [22,124].

Correct the word “preimplantation” and add references 1, 20 and 125, which show XCR in preimplantation embryos in Nanog+ cells (reference 22 shows it during iPSC reprogramming but not in preimplantation embryos).

We agree with the reviewer. The typing mistake has been corrected. The additional important reference has been included, as well as other references that support this point. “In preimplantation mouse embryos, XCR correlates with the expression of the pluripotency gene Nanog [1,2,25,134,135].” lines 252-254.

  1. line 248: erase not erases

 This sentence has been corrected to "Reprogramming somatic cells into iPSCs erases memory...", line 284.

  1. line 582: pluripotency not ply potency 

This has been corrected, line 626.

Reviewer 2 Report

Panda and Pasque review literature on and provide new insights of X chromosome reactivation (XCR). They give an extensive overview of all the stages and mechanisms involved in X chromosome inactivation and XCR and address each of them separately. They end by giving some future perspectives. Overall the manuscript is interesting and suitable for Journal "Cells', but I do have some comments and suggestions that I would like to see addressed prior to publication.

Major comments

The manuscript would be much improved if authors briefly summarized the methods that were used to study each of the mechanisms involved in XCI and XCR (sections 2-13). Some of the sections (6 and 11) contain this information, but it would be good for the audience to make it consistent across all of them. 

Trans and cis factors:

The authors briefly mention trans factors involved in the expression of the Xa. It would be good to introduce readers to trans and cis factors in more details.

A summary figure at the end of the manuscript underlining the gaps in our knowledge of XCR would significantly improve the review.

Minor comments

Figures: Spell out the abbreviations in the figures' legends for clarity. You spell out the abbreviation in the legend of Figure 3, but not in Figure 1 and 2.

Page 1, line 34: remove “compensation”

Page 2, line 61: Xas and not Xa’s, this error should be corrected throughout the text

Page 2, line 71: you should use “naive” and not “naïve” throughout the text

Page 2, line 86: missing dot at the end of the sentence

Page 3, line 100: should be “an RNA cloud”

Page 3, line 101: should be “Figure 1a”

Page 3, line 102: “chromosomes” and not “chromosome”, “inactivate” and not “inactive”

Page 3, line 112: “Figure 1b”

Page 3, line 122: “stable” used twice

Page 4, line 152: “aging” and not “ageing”

Page 4, line 172: rephrase “established role in establishment”

Page 4, lines 172-173: “also” used twice

Page 5, line 188: “extraembryonic” should be used consistently throughout the text 

Page 5, line 216: should be “preimplantation”

Page 5, line 227: “additional” used twice

Page 6, line 248: “erase” and not “erases”

Page 6, line 275: (a) is part of the Figure 2 on the next page

Page 7, line 290: remove “most; “

Page 7, lines 294-297: this sentence is confusing and could be split into two sentences for clarification

Page 7, line 295: This abbreviation was used here for the first time and needs to be spelled out

Page 7, line 299: sometimes you use iPS cells sometimes iPSC. The term should be introduced and abbreviated at the beginning of the manuscript and this abbreviation should be used later on throughout the text

Page 7-8, line 313-314: In the first part of the sentence you use (human ESCs) and in the second human embryonic stem cells (hESCs). once ESCs is introduced you can introduce human ESCs (hESCs) throughout the text 

Page 8, after line 320: there should be only one-line space

Page 8, line 341: “Figure 3” and not “Figure-3”

Page 8, Figure 3: legend should be placed below the figure

Page 10, line 458: “is” and not “if”

Page 10-11, lines 458-460: this sentence needs to be rewritten for clarification

Page 11, line490: IF is introduced here for the first time and should be spelled out

Page 11, line 496: ICM is introduced here for the first time and should be spelled out

Page 11, lines 495-496: the verb is missing in the second part of the sentence

Page 11, line 499: EPI is introduced here for the first time and should be spelled out

Page 11, line 502: would be good to spell out “sc” here

Page 12, line 509: PE is introduced here for the first time and should be spelled out

Page 12, line 520: PGCs is introduced here for the first time and should be spelled out. Before you used progenitor cells so introduce the abbreviation earlier on and use it throughout the text

Page 12, line 522: previously you used “biallelic”

Page 12, line 551: sometimes you use germ line, other times you use germline. Use “germ line” consistently throughout the text

Page 12, line 554: The “hiPSCs” abbreviation was not introduced before

Page 13, line 556: remove one “of”

Page 13, line 558: should be “an RNA cloud”

Page 13, line 562: should be “escapee” 

Page 13, line 582: should be “pluripotency” and not “ply potency”

Page 13, lines 586-587: “from one then from two alleles is a 586 small proportion of cells” seems out of place

Page 13, line 597: remove “then”

Page 14, line 612: you should use “pluripotency TFs” throughout the text after the abbreviation introduction. 

Author Response

Point-by-point Response to the Reviewers Comments:

Manuscript ID: cells-1003543

Reviewer 2 also expressed interest in the manuscript and deemed it suitable for this journal. He/she also had a number of comments and suggestions, which we have addressed.

Comments and Suggestions for Authors

Panda and Pasque review literature on and provide new insights of X chromosome reactivation (XCR). They give an extensive overview of all the stages and mechanisms involved in X chromosome inactivation and XCR and address each of them separately. They end by giving some future perspectives. Overall the manuscript is interesting and suitable for Journal "Cells', but I do have some comments and suggestions that I would like to see addressed prior to publication.

We are thankful to the reviewer for accepting to take the time to go through the review line by line and provide many valuable comments which helped improved the quality of the review.

Major comments

The manuscript would be much improved if authors briefly summarized the methods that were used to study each of the mechanisms involved in XCI and XCR (sections 2-13). Some of the sections (6 and 11) contain this information, but it would be good for the audience to make it consistent across all of them. 

We agree with the reviewer. An effort has been made to mention methods in all the appropriate places throughout the manuscript. [lines [120], [157], [167], [190], [201], [212], [251], [281-283], [319], [322-324], [331-333], [340], [346], [380-383], [437-438], [476], [533-534], [546], [565], [574], [582], [597], [604], [621-623], [631]].

Trans and cis factors:

The authors briefly mention trans factors involved in the expression of the Xa. It would be good to introduce readers to trans and cis factors in more details.

We agree with the reviewer. The sentence has now been rephrased. "Once established, chromatin silencing on the Xi is maintained despite the presence of strong trans factors that promote transcriptional activation of the Xa (transfactors are factors that have the potential to act on both alleles). Chromatin silencing in cis (affecting only the same allele) prevents reactivation by transcription factors and forms stable and heritable gene silencing.", [lines 78-83].

A summary figure at the end of the manuscript underlining the gaps in our knowledge of XCR would significantly improve the review.

A summary figure underlying the gaps in our knowledge of XCR has now been added, page 15.

Minor comments

Figures: Spell out the abbreviations in the figures' legends for clarity. You spell out the abbreviation in the legend of Figure 3, but not in Figure 1 and 2.

We made the changes in line [129-134] and [314-315].

Page 1, line 34: remove “compensation”

""compensation" has been removed, line [34].

Page 2, line 61: Xas and not Xa’s, this error should be corrected throughout the text

We made the changes throughout the manuscript, as suggested.

Page 2, line 71: you should use “naive” and not “naïve” throughout the text

'Naïve' has been changed to 'naive'.

Page 2, line 86: missing dot at the end of the sentence

We made the changes in line [113].

Page 3, line 100: should be “an RNA cloud”

This has been adopted throughout the manuscript.

Page 3, line 101: should be “Figure 1a”

We made the changes for the figures including this one.

Page 3, line 102: “chromosomes” and not “chromosome”, “inactivate” and not “inactive”

Thank you for the comments. This has been changed as per the reviewer’s suggestion.

Page 3, line 112: “Figure 1b”

This has been corrected, line [136].

Page 3, line 122: “stable” used twice

Thank you for noting this. We restructured the sentence, line [148].

Page 4, line 152: “aging” and not “ageing”

"ageing" has been turned into "aging" everywhere except in the title of the Migeon et al. Nature 1988 reference title who used the British spelling of the word.

Page 4, line 172: rephrase “established role in establishment”

We reframed the sentence, line [203-206].

Page 4, lines 172-173: “also” used twice

The sentence has been restructured. Line [203-206].

Page 5, line 188: “extraembryonic” should be used consistently throughout the text

As suggested, it has been now used consistently throughout the manuscript. 

Page 5, line 216: should be “preimplantation”

This also has been adopted throughout the manuscript including here in line 252.

Page 5, line 227: “additional” used twice

"additional" has been deleted, line [264]

Page 6, line 248: “erase” and not “erases”

The correction has been adopted, line [284].

Page 6, line 275: (a) is part of the Figure 2 on the next page

Care has now been taken to avoid Figure splitting across multiple pages.

Page 7, line 290: remove “most; “

'most' has been removed', line [327]

Page 7, lines 294-297: this sentence is confusing and could be split into two sentences for clarification

As suggested, the sentence has now been split into two.

Page 7, line 295: This abbreviation was used here for the first time and needs to be spelled out

We disagree with the reviewer. ESCs is already defined on line [58].

Page 7, line 299: sometimes you use iPS cells sometimes iPSC. The term should be introduced and abbreviated at the beginning of the manuscript and this abbreviation should be used later on throughout the text

We totally agree with the reviewer and we changed this in the entire manuscript.

Page 7-8, line 313-314: In the first part of the sentence you use (human ESCs) and in the second human embryonic stem cells (hESCs). once ESCs is introduced you can introduce human ESCs (hESCs) throughout the text 

As requested, we have made the valuable suggested change, line [350-353].

Page 8, after line 320: there should be only one-line space

This has now been corrected, line [357-358].

Page 8, line 341: “Figure 3” and not “Figure-3”

Figure-3 was changed to Figure 3, as suggested, line [378].

Page 8, Figure 3: legend should be placed below the figure

Care has now been taken to position legends below figures.

Page 10, line 458: “is” and not “if”

We considered the typographic error and made the change. Line [500].

Page 10-11, lines 458-460: this sentence needs to be rewritten for clarification

We have now rewritten the sentence, which now reads " First, in somatic cells, Xist deletion is not sufficient to enable TFs involved in transcriptional activation on the Xa to bind to the Xi and induce transcription [112,113]", line [500-502].

Page 11, line490: IF is introduced here for the first time and should be spelled out

IF was introduced earlier line [167].

Page 11, line 496: ICM is introduced here for the first time and should be spelled out

We removed ICM from the manuscript and used preimplantation epiblast.

Page 11, lines 495-496: the verb is missing in the second part of the sentence

The sentence has been rewritten and now reads " Moreover, there is no imprinted XCI in humans [183], hence XCR does not take place in the human preimplantation epiblast. ", line [538-539].

Page 11, line 499: EPI is introduced here for the first time and should be spelled out

EPI is now spelled out everywhere.

Page 11, line 502: would be good to spell out “sc” here

Sc has now been spelled out, as requested "Single-cell". Line [545].

Page 12, line 509: PE is introduced here for the first time and should be spelled out

PE is introduced previously in the page number 2 line [62].

Page 12, line 520: PGCs is introduced here for the first time and should be spelled out. Before you used progenitor cells so introduce the abbreviation earlier on and use it throughout the text

PGCs are primordial germ cells and it is introduced and defined in section 4 line [269].

Page 12, line 522: previously you used “biallelic”

 We changed  “biallelic” to “bi-allelic” first in line [118] and throughout the manuscript.

Page 12, line 551: sometimes you use germ line, other times you use germline. Use “germ line” consistently throughout the text

We now use 'germline' everywhere.

Page 12, line 554: The “hiPSCs” abbreviation was not introduced before

The abbreviation is now introduced in line [X].

Page 13, line 556: remove one “of”

Thank you for noticing the typing error. We made the change on line [600].

Page 13, line 558: should be “an RNA cloud”

"a RNA" cloud has now been edited into "an RNA cloud" throughout the manuscript.

Page 13, line 562: should be “escapee” 

We changed into “escapee”.

Page 13, line 582: should be “pluripotency” and not “ply potency”

The correction has been made, line [626].

Page 13, lines 586-587: “from one then from two alleles is a 586 small proportion of cells” seems out of place

The sentence has been edited, line [628-631].

Page 13, line 597: remove “then”

'then' was changed to 'and', line [642].

Page 14, line 612: you should use “pluripotency TFs” throughout the text after the abbreviation introduction.

We now use "pluripotency TFs'' throughout the text after the abbreviation is introduced, as requested.

Reviewer 3 Report

In the present manuscript Panda and Pasque review our current knowledge and the most recent insights into X chromosome reactivation (XCR) in mammals. XCR , the process by which X chromosome silencing is reversed, is an important process and can be extremely informative on general mechanisms that regulate chromatin structure and function in mammals. The development of the induced PSC field and the most recent establishment of single cell transcriptomic and epigenetic profiles has further stimulated research on this topic.

The review is quite extensive and well-written and it is timely and of interest for readers of cells, despite it lacks a bit of synthesis. To further promote reading and understanding to non experts of the field I would suggest the authors to be more synthetic and avoid some redundancies in the text.

Apart form this I have just a few minor comments/edit

Lane 172-173: the word also is used twice in the same sentence. I would avoid that

Lanes 294-298: sentence is not clear. I would divide it into two or correct it

I would also briefly describe the Xic locus as authors extensively refer to it and its regulation. Same applies to Tsix.

Please also briefly define escapee genes (lane 300 first time I guess)

Lane 388: sentence is not clear….interaction within decrease TAD-E (within TAD-E decrease)

Lane 458: please correct if to is. Also the sentence does not read very well

Lane 586 please correct is to in. also the sentence is very long.

Author Response

Point-by-point Response to the Reviewers Comments:

Manuscript ID: cells-1003543

Reviewer 3 appreciated the importance and value of studies of X chromosome reactivation to inform general mechanisms that regulate chromatin structure and function in mammals. He/She found the review "extensive and well-written and [...] timely and of interest for readers of cells". This reviewer also felt that the review would be improved with a bit more of synthesis and made suggestions for improvement, which we have now addressed.

Comments and Suggestions for Authors

In the present manuscript Panda and Pasque review our current knowledge and the most recent insights into X chromosome reactivation (XCR) in mammals. XCR, the process by which X chromosome silencing is reversed, is an important process and can be extremely informative on general mechanisms that regulate chromatin structure and function in mammals. The development of the induced PSC field and the most recent establishment of single cell transcriptomic and epigenetic profiles has further stimulated research on this topic.

The review is quite extensive and well-written and it is timely and of interest for readers of cells, despite it lacks a bit of synthesis. To further promote reading and understanding to non experts of the field I would suggest the authors to be more synthetic and avoid some redundancies in the text.

We are grateful to the reviewer for the extensive effort of going through the review line by line and providing many constructive comments which immensely help improving the quality of the review. In an attempt to be more synthetic, we have made an effort to remove some redundancies in the text and adopted additional changes aimed at making the review more accessible to non-experts.

Apart form this I have just a few minor comments/edit

Lane 172-173: the word also is used twice in the same sentence. I would avoid that

Thank you for pointing this out. This has been corrected, line [205]. Efforts to avoid this type of duplication have been made.

Lanes 294-298: sentence is not clear. I would divide it into two or correct it

We agree with the reviewer. We have now divided the sentence into two, line [331-335].

I would also briefly describe the Xic locus as authors extensively refer to it and its regulation. Same applies to Tsix.

The Xic has been briefly described in line [126-127].

We have now included an additional line about Tsix, line 94. 

Please also briefly define escapee genes (lane 300 first time I guess)

As suggested, we have now defined escapee genes. Line [116-118].

-Lane 388: sentence is not clear….interaction within decrease TAD-E (within TAD-E decrease)

The sentence has now been clarified, line [430-432].

Lane 458: please correct if to is. Also, the sentence does not read very well

The correction has been made, line [500-502].

Lane 586 please correct is to in. also the sentence is very long.

The correction has been made and the sentence has been split into two.

Round 2

Reviewer 1 Report

The authors have now implemented all suggested changes by the reviewers and the manuscript is now be suitable for publication.